# Systematic Functional Annotation Workflow for Insects

**DOI:** 10.3390/insects13070586

**Published:** 2022-06-27

**Authors:** Hidemasa Bono, Takuma Sakamoto, Takeya Kasukawa, Hiroko Tabunoki

**Affiliations:** 1Laboratory of Bio-DX, Genome Editing Innovation Center, Hiroshima University, 3-10-23 Kagamiyama, Higashi-Hiroshima City 739-0046, Japan; 2Laboratory of Genome Informatics, Graduate School of Integrated Sciences for Life, Hiroshima University, 3-10-23 Kagamiyama, Higashi-Hiroshima City 739-0046, Japan; 3Institute of Global Innovation Research, Tokyo University of Agriculture and Technology, 3-5-8 Saiwai-cho, Fuchu, Tokyo 183-8509, Japan; tsakamoto@go.tuat.ac.jp (T.S.); h_tabuno@cc.tuat.ac.jp (H.T.); 4Department of Science of Biological Production, Graduate School of Agriculture, Tokyo University of Agriculture and Technology, 3-5-8 Saiwai-cho, Fuchu, Tokyo 183-8509, Japan; 5RIKEN Center for Integrative Medical Sciences, 1-7-22 Suehiro-cho, Tsurumi-ku, Yokohama 230-0045, Japan; takeya.kasukawa@riken.jp

**Keywords:** functional annotation, RNA sequencing, transcriptome assembly, stick insect, silkworm

## Abstract

**Simple Summary:**

The functions of all genes encoded in the genome should be studied for genome editing. Genome editing technology can speed up insect research for the functional analysis of genes. Our knowledge about the functional information of genes is still incomplete currently, while the genome sequencing of an organism can be completed. The functional information has been annotated based solely on the information that has been obtained from the results of previous biological research. However, this information will be important in determining the target genes for genome editing. In particular, it is very important that this information is in machine-readable form because computer programs mainly parse this information for the understanding of biological systems. In this paper, we describe a workflow-based method for annotating gene functions in insects that makes use of transcribed sequence information as well as reference genome and protein sequence databases. Using the developed workflow, we annotated the functional information of the Japanese stick insect and silkworm, including gene expression as well as sequence analysis. The functional annotation information obtained by the workflow will greatly expand the possibilities of entomological research using genome editing.

**Abstract:**

Next-generation sequencing has revolutionized entomological study, rendering it possible to analyze the genomes and transcriptomes of non-model insects. However, use of this technology is often limited to obtaining the nucleotide sequences of target or related genes, with many of the acquired sequences remaining unused because other available sequences are not sufficiently annotated. To address this issue, we have developed a functional annotation workflow for transcriptome-sequenced insects to determine transcript descriptions, which represents a significant improvement over the previous method (functional annotation pipeline for insects). The developed workflow attempts to annotate not only the protein sequences obtained from transcriptome analysis but also the ncRNA sequences obtained simultaneously. In addition, the workflow integrates the expression-level information obtained from transcriptome sequencing for application as functional annotation information. Using the workflow, functional annotation was performed on the sequences obtained from transcriptome sequencing of the stick insect (*Entoria okinawaensis*) and silkworm (*Bombyx mori*), yielding richer functional annotation information than that obtained in our previous study. The improved workflow allows the more comprehensive exploitation of transcriptome data and is applicable to other insects because the workflow has been openly developed on GitHub.

## 1. Introduction

Genome sequencing is becoming common practice in insect research. As of May 2022, the genomes of around 3000 insect species have been decoded and registered in the genome section of the NCBI Datasets [1]. In addition, long-read technology is further accelerating the pace of insect genome sequencing [2].

However, some insects have genomes larger than that of humans, further complicating the difficult process of whole-genome sequencing. As an alternative, transcriptome sequencing using next-generation sequencing technology, also termed RNA sequencing (RNA-Seq), provides a powerful tool for evaluating non-model species such as large-genome-size insects [3]. In particular, this strategy can efficiently identify tens of thousands of possible genes in a specific tissue by assembling tens of millions of reads. The sequences are then assembled for the identification of transcriptional units. Systematic sequence similarity searches against non-redundant sequence sets and BLAST2GO searches [4] are often employed for the functional inference of such transcriptomes to be utilized in gene set enrichment analyses. Moreover, a new method termed Seq2Fun that does not require transcriptome de novo assembly was recently developed for the rapid functional profiling of RNA-Seq data for non-model organisms [5]. Nevertheless, the output of such analyses is reliant upon the comprehensiveness of the available datasets and their functional annotation. 

Traditionally, reference data have been maintained as research community databases (DBs), such as FlyBase for fruit flies [6]. In turn, such communities spearheaded the development of Gene Ontology (GO) [7], which provides machine-interpretable annotations incorporating human-readable descriptions. These annotations for genomic research, including GO, are provided by Ensembl and cover a variety of vertebrates [8]. For insects, Ensembl Metazoa in Ensembl Genomes integrates the community annotations of individual insects [9]. Nevertheless, it is unable to keep pace with the increase in insect genome sequencing, including currently available genome and transcriptome data. 

Concurrently, we developed a functional annotation pipeline in non-model insects for the microarray-based transcriptome analysis of a unique mutant of the silkworm *Bombyx mori op* [10]. Subsequently, we also utilized this pipeline for comparative studies with humans and flies [11,12]. In particular, by assigning human gene identifiers (IDs) to the silkworm genes, we could facilitate the reconstruction of pathway DBs, which were originally developed for the analysis of model organisms [10,13]. However, such prior studies have almost exclusively been limited to protein coding sequences, as functional annotation pipelines for transcriptome sequences were originally designed for mammalian transcriptomes [14] produced by sequencing cDNAs including expressed sequence tags (ESTs) using Sanger sequencers in the Functional Annotation of Mammalian genomes (FANTOM) project [15,16]. Although functional annotation for non-coding RNAs (ncRNA) was considered in the pipeline for FANTOM3 [17], this feature was ultimately not incorporated because the reference information and biological knowledge regarding ncRNAs were not yet sufficient [18]. 

As transcriptome sequencing becomes popular, many groups run pipelines of their own, with the information regarding the transcription units from various studies being reported on a study-by-study basis. Moreover, the transcript assemblies are often not registered in a public DB, even though it is recommended that such data be deposited in the Transcriptome Shotgun Assembly (TSA) sequence database under the International Nucleotide Sequence Database Collaboration (INSDC) [19,20], as the data need to be annotated uniformly to facilitate reuse in other analyses. However, the functional information of the currently registered transcription units is not sufficiently rich to support functional analysis and almost never annotated once archived, even if it is available. Thus, the functional annotation of transcriptomes in a manner that can be integrated with datasets from other groups including the data in public DBs is urgently required.

Therefore, in this study, we evaluated the functional annotation of transcripts from assembled transcriptomes to establish a functional annotation workflow for insect transcriptomes by updating the previously developed functional annotation pipeline for silkworm [10]. Then, we developed a systematic functional annotation workflow for insect research termed “Fanflow4Insects”. Fanflow4Insects enables the automatic annotation of sequences to exploit the exponential increase in genome and transcriptome information. Finally, Fanflow4Insects was tested using not only the reference transcriptome of silkworm (*Bombyx mori*) but also the transcriptomes of the Japanese stick insect (*Entoria okinawaensis*), including the newly sequenced transcriptome of the fat body.

## 2. Materials and Methods

### 2.1. Programs Comprising the Workflow

All scripts that form the workflow described below are freely accessible from the Systematic Analysis for Quantification of Everything (SAQE) [21] code repository at GitHub [22], along with the detailed parameters.

For transcriptomes without a reference genome sequence, we utilized Trinity (v2.12.0) with paired-end parameters in the Docker environment (trinityrnaseq/trinityrnaseq:2.12.0) to construct de novo transcriptome assemblies [23]. 

Protein coding sequences within transcriptome sequences produced by de novo assembly or by mapping the reference genome sequence were predicted using TransDecoder (v5.2.0) [24]. The predicted and then translated protein sequences were subsequently compared via successive execution of the GGSEARCH program (ggsearch36; v36.3.8g) with ‘-d1 -m 10 -E 0.1′ parameters in the FASTA package [25] against the protein reference sequence DBs described in Table 1. The protein domain search program (hmmscan) in the HMMER package [26] was used for the predicted protein sequences. Pfam (v35.0) was used as the protein domain DB [27].

The transcriptome sequences were then used for annotation as ncRNA sequences. Nucleotide BLAST (blastn; v2.6.0+) [28] was used to detect sequence similarity to known ncRNA sequences in Ensembl (human) [8] and Ensembl Genomes (fly) [9] DBs. The program (cmscan) in the Infernal: inference of the RNA alignments package (v1.1.4) [29], which was developed for searching nucleotide sequence DBs for RNA structure and sequence similarities, was used to annotate transcript sequences against Rfam (v14.7), utilized as the RNA family DB [30].

A program (align_and_estimate_abundance.pl) in the Docker container for Trinity (trinityrnaseq/trinityrnaseq:2.11.0) was used to estimate the abundance of RNA-Seq reads [23]. Salmon (v1.0.0) [31] was selected to quantify the transcripts to produce the expression matrix for each organism.

### 2.2. Insects and Sample Collection

The stick insects *E. okinawaensis* were obtained from Amami-Ohshima in Kagoshima, Japan, in 2011 and Ishigaki Island in Okinawa, Japan, in 2013, as described in our previous study [32]. Three biological replicates of the fat body were dissected from adult *E. okinawaensis*. These tissues were stored at −80 °C until use. The fat body samples were weighed, homogenized with lysis buffer from a PureLink^®^ RNA extraction kit (Thermo Fisher Scientific Inc., Valencia, CA, USA), and then centrifuged at 13,000× *g* for 10 min. The supernatants were then collected and processed for RNA purification according to the manufacturer’s instructions. Purified total RNA samples (1 µg each) were processed for RNA sequencing.

### 2.3. RNA Sequencing

RNA quality was assessed using a Bioanalyzer 2100 (Agilent Technologies, Santa Clara, CA, USA). Libraries for cDNA sequencing were constructed using the Illumina TruSeq v2 kit (Illumina Inc., San Diego, CA, USA), according to the manufacturer’s protocol, and 100 bp paired-end sequencing was then performed on the HiSeq 2500 platform (Illumina). Using Trim Galore! (v0.4.4) [33], low-quality bases and adapter sequences were trimmed, and the data quality was visualized and then confirmed. The raw RNA sequence reads for biological triplicates of *E. okinawaensis* fat body samples were deposited in the Sequence Read Archive (SRA) under accession ID DRA013458 (DRR346171, DRR346172, and DRR346173). 

## 3. Results

### 3.1. Functional Annotation Workflow for Insects

We first evaluated rules for the functional annotation of transcripts. In our previous studies, we only focused on translated protein sequences [12,32]. In the present study, we also aimed to annotate sequences without coding regions using ncRNA references, and quantified expression information from RNA-Seq as well as protein coding sequences ascertained based on sequence similarity. For transcripts with no sequence similarity, quantified expression values were used for annotation (Figure 1). The functional annotation workflow described below, being developed especially for insects, was termed “Fanflow4Insects”.

#### 3.1.1. Functional Annotation of Coding Sequences

Using predicted protein sequences translated from each assembled transcriptome, we performed systematic sequence similarity searches against the predicted protein sequence datasets of functionally well-annotated organisms, including human (*Homo sapiens*), mouse (*Mus musculus*), nematode (*Caenorhabditis elegans*), and fruit fly (*Drosophila melanogaster*). Because local alignment is not appropriate for evaluating the overall percentage similarity between protein sequences, we adopted global alignment as an alternative as insects are far distant from well-annotated reference organisms. The program GGSEARCH in the FASTA package [25] was used for implementation of the global alignment search, and organism-specific homology searches by GGSEARCH were executed against the four functionally well-annotated organisms described above, followed by a search against UniProtKB [34]. A corresponding functional description of the top hit in the reference organism or UniProtKB was annotated as “homolog” (Figure 2).

If no homologs could be found, the search result of HMMSCAN against Pfam was referenced [27] (Figure 2). According to the protein domain hits, protein domain information was annotated if present (example: “zinc finger containing protein”); otherwise, only “hypothetical protein” was used. The sources of the reference DBs used in the functional annotation are summarized in Table 1.

#### 3.1.2. Functional Annotation of Non-Coding Sequences

For sequences in which TransDecoder was unable to find any protein coding region (i.e., >100 amino acids (aa) in length by default), functional information of the ncRNAs was annotated based on nucleotide Basic Local Alignment Search Tool (BLAST) search against ncRNA reference sequences (Figure 2). For insect functional annotation, reference ncRNA sequences of the four functionally well-annotated organisms (*H. sapiens*, *M. musculus*, *C. elegans*, and *D. melanogaster*) were used in Fanflow4Insects (Table 1).

If no functional information could be assigned, then the RNA structure and sequence similarity against Rfam [29] were assessed using the CMSCAN program in the Infernal package [28] (Figure 2). If any information regarding the sequence was located, the RNA-related information in DB hits was annotated; otherwise, the sequence was annotated as an “unclassifiable transcript”.

#### 3.1.3. Functional Annotation by Quantified Expression Information

Transcript expression was quantified based on the assembled transcriptome and RNA-Seq raw reads. The matrix of gene expression was generated by pasting columns. Using the expression matrix, we were able to annotate gene expression features for all genes. 

Although the threshold for the judgment remains to be calibrated, genes with specific expression were then annotated (Figure 3). If a transcript was expressed in all tissues, it was annotated as “constitutive expression”. Alternatively, if expressed only in, e.g., the fat body, it was annotated as “fat body-specific expression”. Practical examples of annotation based on expression information are described in the following section.

### 3.2. Case Study for Fanflow4Insects

Practical applications of Fanflow4Insects for the Japanese stick insect *Entoria okinawaensis* and silkworm *Bombyx mori* are described below.

#### 3.2.1. Functional Annotation of the Japanese Stick Insect Transcriptome Using Fanflow4Insects

In addition to three biological replicates of the *E. okinawaensis* midgut sample, which we previously reported and deposited in SRA under DRA007226 (DRR148118, DRR148119, and DRR148120) [32], three biological replicate reads of the *E. okinawaensis* fat body (DRR346171, DRR346172, and DRR346173) were used for the functional annotation by Fanflow4Insects. Transcriptome completeness analysis with BUSCO v5.3.2 using the insecta_odb10 dataset (2020-09-10) with default settings (transcriptome mode) showed 97.1% completeness (43.5% single-copy and 53.6% duplicated) for transcriptomes from the fat body and midgut of *E. okinawaensis*, while it showed 93.0% completeness (47.9% single-copy and 45.1% duplicated) for that only from the midgut of *E. okinawaensis* [35].

Of 311,357 transcripts, 18.5% of these (57,732 transcripts) could be translated into 68,162 predicted protein sequences with more than 100 amino acids in length. Some transcripts were predicted to encode more than one protein sequence. These protein coding sequences were annotated by systematic sequence similarity and protein domain searches (Table 2). In brief, 55,068 predicted protein sequences (80.8%) had homologs and 58,876 of these (86.4%) had some degree of sequence similarity information including protein domains. In total, 9286 predicted protein sequences could not be assigned to any functional information from sequence similarity and domain searches in protein level and were thus annotated as “hypothetical proteins”.

The remaining 253,625 sequences comprised transcripts without protein coding regions. Of these, only 185 sequences had sequence similarity to known ncRNA sequences. Moreover, 264 sequences could be assigned based on a CMSCAN search against Rfam DB, whereas 253,176 sequences were annotated as an “unclassifiable transcript”. These functional annotation results from sequence information for *E. okinawaensis* are summarized graphically in Appendix A.

For the hypothetical protein and the unclassifiable transcripts with no sequence similarity, quantified expression values were used for annotation. Functional information could be inferred and annotated based on the expression matrix generated from sequence counts (Figure 1). Table 3 summarizes the numbers of sequences annotated as “hypothetical protein” and “unclassifiable transcript” and the corresponding expression annotations. Only a small fraction of sequences, 2.7% (156) of “hypothetical proteins” and 1.5% (3860) of “unclassifiable transcripts”, had no expression information (Table 3).

The assembled transcriptomes and the estimated transcript abundances were deposited in the TSA Sequence database under accession IDs ICSG01000001–ICSG01311357 and the Genomic Expression Archive (GEA) under accession ID E-GEAD-476. Full functional annotation and transcript descriptions generated by Fanflow4Insects were deposited in figshare to be reused in stick insect research (https://doi.org/10.6084/m9.figshare.19368110.v1 (accessed on 23 June 2022)).

#### 3.2.2. Functional Annotation of Silkworm Using Fanflow4insects

We also evaluated the reference transcriptome data of silkworm *B. mori* as a lepidopteran insect with biological and industrial importance [12]. Corresponding functional annotation is available from the Life Science Database Archive [36]; however, only BLAST top hits for human and fly are listed. We therefore applied Fanflow4Insects to silkworm using the assembled transcriptomes (TSA IDs: ICPK01000001–ICSG01051926) and estimated abundance of transcripts (GEA ID: E-GEAD-315). Transcriptome completeness analysis with BUSCO v5.3.2 for the *B. mori* transcriptome by the lepidoptera_odb10 dataset (2020-05-08) yielded 98.7% completeness (41.0% single-copy, 57.7% duplicated) [35]. Of 51,926 transcripts, 78.1% (40,555 transcripts) could be translated into 45,719 predicted protein sequences of length > 100 aa. These protein coding sequences were annotated via systematic sequence similarity and protein domain searches, with 38,107 predicted protein sequences (83.4%) able to be annotated with homologs (Table 4). Moreover, protein domains could be assigned to 5398 predicted protein sequences for the remaining 7612 predicted protein sequences without homologs, resulting in only 2214 predicted protein sequences (4.8%) being annotated as a “hypothetical protein” (Table 4).

The remaining 11,371 sequences comprised transcripts without protein coding regions. Of these, only 45 sequences had sequence similarity to known ncRNA sequences. Moreover, 300 sequences could be assigned based on a CMSCAN search against Rfam DB. Finally, 11,026 sequences were annotated as an “unclassifiable transcript”. These functional annotation results from sequence information for *B. mori* are summarized graphically in Appendix A.

The estimated abundances of transcripts in the expression matrix (GEA ID: E-GEAD-315) were also used for annotation. Table 5 summarizes the numbers of sequences annotated as “hypothetical protein” and “unclassifiable transcript”, together with the corresponding expression annotations. Unlike the results for *E. okinawaensis*, in which data from only two tissues were available, the abundance of transcripts in six tissues, notated as transcripts per million (TPM), was available for *B. mori*. A small number of sequences were judged to be tissue-specific because only the genes with expression in specific tissues (TPM > 0) and no expression (TPM = 0) in other tissues were considered to be tissue-specific. In summary, 3.0% (44) of 1486 transcripts carrying the “hypothetical protein” annotation and 5.4% (693) of 12,845 annotated as an “unclassifiable transcript” could be annotated with strong expression features, although the function of these transcripts could not be inferred based only on sequence similarity information (Table 5).

The functional annotation described above, including the transcript descriptions generated by Fanflow4Insects along with the expression annotation, has been deposited in figshare for use in silkworm research (https://doi.org/10.6084/m9.figshare.19368137.v1 (accessed on 23 June 2022)).

#### 3.2.3. Comparison of ncRNAs between Japanese Stick Insect and Silkworm

A comparative analysis of ncRNA transcripts was performed to utilize the output of Fanflow4Insects. Notably, the functional annotation to ncRNA references (human and fruit fly) by Fanflow4Insects provided transcript IDs for *E. okinawaensis* and *B. mori* along with corresponding ncRNA IDs from reference organisms for use in this comparison (Figure 4).

Whereas few transcripts were functionally annotated based on existing ncRNAs in the public database, several functionally annotated transcripts were identified in the intersection of *E. okinawaensis* and *B. mori*; the corresponding transcript IDs in the reference organisms (human and fruit fly) are listed in Table 6. All *E. okinawaensis* transcripts annotated and then listed in Table 6 had more than one counterpart transcript in *B. mori* by BLASTN search. We also evaluated the corresponding functional annotation derived from the expression of these transcripts, revealing that most transcripts exhibit constitutive expression, whereas several show midgut-specific expression in *E. okinawaensis*; in contrast, all transcripts are constitutively expressed in *B. mori*.

## 4. Discussion

Next-generation sequencing has facilitated the decoding of the genomes of numerous insects along with their transcript sequences. However, the biological interpretation of these sequences remains a primary bottleneck of transcriptome analysis. An important first step is functional annotation, which serves as, e.g., an important clue for selecting genome editing targets. Toward this end, we have had a long-term interest in providing functional annotation, starting with the annotation of enzymatic genes using the Gene Function Identification Tool (GFIT) in bacteria [37]. More recently, we have contributed to the functional annotation pipeline in the FANTOM project to provide the scientific community with an accurate functional annotation of mouse cDNA [15,16,17].

Currently, BLAST2GO is often used for the functional annotation of non-model organisms [4]; however, BLAST2GO is a commercial application and a considerable delay exists before results are obtained. Recently, Seq2fun has been used to predict the function of the total set of genes encoded in a genome at very high speed [5]. However, Seq2fun does not generate sequences for each gene, making it unusable for applications such as individual transcriptome analysis. Thus, it is not possible to trace how the functional annotation was assigned to each gene using Seq2fun. We consider that traceable functional annotation supported by concrete evidence is crucial in biological studies. In addition, these tools cannot be customized to the circumstances of each organism. To address these issues, we therefore developed Fanflow4Insects.

In our previous study for annotating the *E. okinawaensis* midgut transcriptome, we performed a systematic sequence similarity search using translated protein sequences against the predicted protein sequences of well-annotated model organisms using a locally installed NCBI protein BLAST program [32], which provided sufficient speeds to obtain results in a reasonable time. However, although BLAST reports the local alignment of the query sequence and DB hit sequence [28], the local alignment is not appropriate for the present purposes because the overall sequence identity and sequence similarity between protein sequences cannot be determined. Therefore, in the present study, we re-evaluated the program to be applied for the project. The program FASTY in the FASTA package [25], previously utilized in the FANTOM project [14,18], compares a DNA sequence to a protein sequence DB, translating the DNA sequence in three forward (or reverse) frames and allowing frameshifts. However, despite dramatic improvements in computation technology, FASTY still requires considerable computing power and is difficult to use for practical functional annotation. Considering that next-generation sequencing technology using “sequence by synthesis” with Illumina sequencers is much more accurate as compared to capillary sequencing using the Sanger method, and nucleotide deletions are less likely to occur in the final sequences, we proposed to utilize global rather than local alignment in the present study. Global alignment is also more appropriate because several well-conserved alignment fragments are apparent upon the alignment of insects and model organisms, while insects are far distant from the well-annotated reference organisms. We found that the GGSEARCH program in the FASTA package is most suitable for the implementation of global alignment searches in a local installation environment. In particular, GGSEARCH calculates sequence identity and similarity, which allows biologists to intuitively evaluate the homology between two sequences, using global alignment, while providing similar speed to BLAST. However, because of the evolutionary distance between insects and well-annotated model organisms, we set a lower E-value threshold of 0.1 in the systematic sequence similarity search by GGSEARCH against genes from well-annotated organisms (human, mouse, *C. elegans*, and *D. melanogaster*) followed by UniProtKB [34] (Figure 2). By this procedure, the functional annotation of genes was dramatically improved by using well-annotated species protein sequence datasets compared with the use of only a non-redundant dataset (nr), where so many genes tend to be annotated as “hypothetical proteins” because gene descriptions of related organisms are frequently assigned. The use of silkworms and Japanese stick insects as well as parasitic wasps *Copidosoma floridanum* made this feature suitable for the functional annotation of insect species [38,39].

In practice, more than 80% of protein sequences for *E. okinawaensis* and *B. mori* could be assigned homologs using GGSEARCH (Table 2 and Table 4). In our previous study, we showed that approximately 58% of *B. mori* genes had human homologs [11]. In the present study, we showed that 68.6% of *B. mori* protein sequences could be assigned human or mouse homologs (Table 4). We believe that the increase in percentage directly reflects advances in our knowledge of reference sequences.

Although many DBs have been developed for protein domains, Pfam [27], which describes protein domains using the Hidden Markov Model (HMM), is often used. We therefore included the program HMMSCAN in the HMMER package [26], available as a program to search Pfam, in the workflow. Notably, we obtained functional information for numerous sequences for which no sequence similarity to the reference protein sequence was identified using protein domain assignment. In particular, some protein domain information could be assigned for 3808 sequences in *E. okinawaensis* and 5398 sequences in *B. mori* (Table 2 and Table 4), which allows the function of the sequence to be inferred despite the lack of similarity to a known protein sequence. Overall, this strategy allowed 95.2% of protein sequences in *B. mori* to be annotated, with only 4.8% (2214 sequences) being left unannotated. These findings indicate that sequence similarity and protein domain searches with proper reference databases (Table 1) can serve as a powerful tool for functional annotation. 

The accumulation of ncRNA reference sequences in model organisms facilitates sequence-level comparisons; thus, we incorporated ncRNAs into the workflow. Toward this end, we utilized nucleotide BLAST, as sequences are not as conserved at the nucleotide sequence level as at the protein sequence level. Therefore, although few assignments were obtained, they were useful to infer functionality. In particular, knowledge regarding RNA domains has been accumulated in a DB called Rfam [30], with RNA domains able to be searched using the program CMSCAN in the Infernal package [29]; thus, this program was also incorporated into the workflow. We anticipate that although only a few sequences were annotated from ncRNA DB and Rfam for *E. okinawaensis* and *B. mori* (Appendix A); additional sequences will be annotated in the future as ncRNA reference sequences and RNA domain information become more complete. While we utilized the CMSCAN program to search against Rfam DB in the current version of Fanflow4Insects, the functional annotation of ncRNA should be improved in a coming version of Fanflow4Insects to draw functional inference from existing databases. While the current target of Fanflow4Insects is to annotate functional information from public databases for typical RNAseq reads of insect species, one should be careful about the information of RNAseq protocols in the annotation of ncRNAs.

Performing functional annotation based on gene expression is challenging, encompassing numerous problems such as how to determine the threshold and the inability to detect weak expression unless the RNA-Seq reads are sufficiently deep [40]. However, gene expression can provide useful information for inferring the function of the sequence. Toward this end, information from derived libraries has been previously used in the analysis of ESTs [15,16,17]. More recently, such information has been replaced by quantified expression values derived from RNA-Seq, which can be used for functional annotation, together with the level of expression in other tissues. Accordingly, we also integrated quantified expression values for providing functional annotation. In the present study, sequences that were expressed only in a specific tissue were annotated as showing tissue-specific expression (Table 3 and Table 5). Notably, as quantified expression data were available for only two tissues in *E. okinawaensis*, multiple sequences were judged as tissue-specific. This allowed us to add functional annotation based on expression to sequences previously annotated as “hypothetical protein” or “unclassifiable transcript” (Table 3). In comparison, as data for six tissues were available in *B. mori*, only a few sequences exhibited tissue-specific expression and could accordingly be assigned functional annotation (Table 5). Moreover, the functional annotation to ncRNA references in Fanflow4Insects allowed the comparative analysis of ncRNA transcripts (Figure 4). In conjunction with the expression-based functional annotation, the comparative use of Fanflow4Insects data could therefore readily identify candidate genes for the future evaluation of, e.g., insects with distinct phenotypes (Table 6). These results indicated that even sequences for which no information was obtained from sequence similarity were able to provide useful functional information.

For enrichment analysis, it is especially important to render the data not only human-readable with regard to transcript description but also machine-readable using GO and other functional annotations. Although originally developed by fly geneticists, GO is now widely used in various model organisms [7]. In our previous study for annotating the *E. okinawaensis* midgut transcriptome, enrichment analyses were performed by assigning transcripts of *E. okinawaensis* to those of *D. melanogaster* because no original GO annotations were available for *E. okinawaensis* [32]. The next target is to annotate GO terms to newly sequenced organisms and utilize these for enrichment analysis. Currently, GO annotation applies only for protein coding sequences; however, we speculate that annotation using a controlled vocabulary including that of GO will be incorporated for the functional annotation of both proteins and transcripts because it affords machine readability. As the genome sequences of more organisms become available, functional annotation of the genes encoded therein will be required, together with GO and protein domain annotation to, e.g., play an active role in selecting which genes to edit for genome editing.

Following the Findable, Accessible, Interoperable, Re-usable (FAIR) principle [41], raw RNA-Seq data and hopefully assembled transcriptome sequences should be registered in the public DB as well as to reproduce data analysis. Our data regarding RNA-Seq reads, transcriptome assembly, and quantified expression values are archived in the public DBs Sequence Read Archive (SRA), TSA, and GEA, respectively. A workflow that integrates these elements is very important and therefore should be re-usable with proper interoperability. Developed openly on GitHub, it can be easily cloned from GitHub repositories. Moreover, specific tools including GGSEARCH can easily be introduced through the bioconda framework, even if the user is not a bioinformatician [42]. All scripts to run the program described in the present study are available on Github [22]. The program is not fully automated as it is expected to be customized for actual use. In particular, gene expression information will vary with each insect species and data acquisition situation. Moreover, the calculation of correspondence relationships with closely related insects for which sequence information is already available will also vary depending on the situation. Such construction of individual workflows according to the available information represents the key to digital transformation while brand new functions in target insects cannot be assigned.

## 5. Conclusions

We have developed a functional annotation workflow for insects, termed “Fanflow4Insects”. Fanflow4Insects has been openly developed on GitHub, and is freely accessible. Fanflow4Insects was tested for functional annotation of the Japanese stick insect *E. okinawaensis* and silkworm *B. mori*. In conjunction with the functional annotation derived from expression, the data from Fanflow4Insects can be applicable to the comparative study of insects with distinct phenotypes. 

## Figures and Tables

**Figure 1 insects-13-00586-f001:**
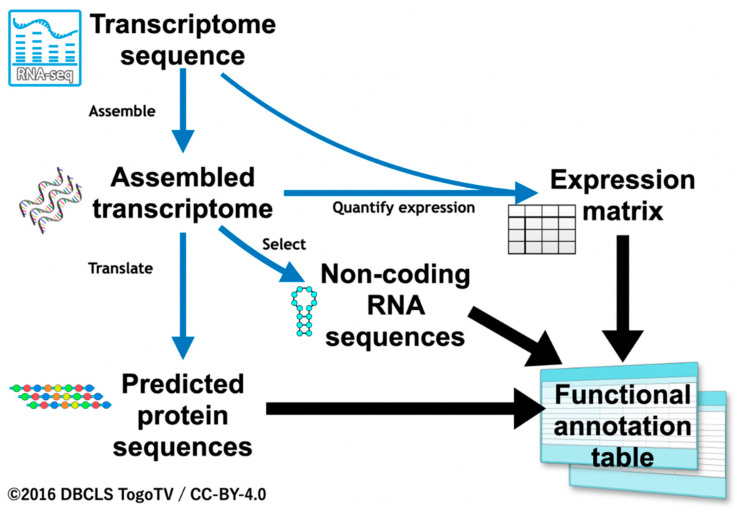
Overview of the annotation workflow, Fanflow4Insects.

**Figure 2 insects-13-00586-f002:**
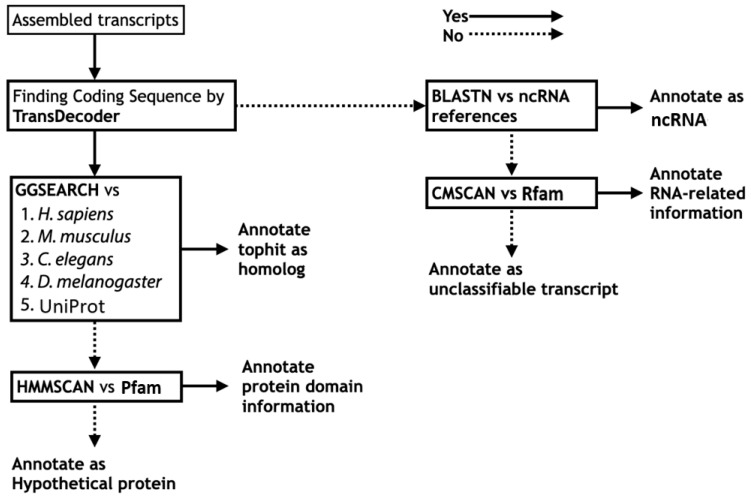
Annotation workflow for transcript description from sequence information. Left half for protein sequence level annotation; right half for nucleotide sequence level annotation.

**Figure 3 insects-13-00586-f003:**
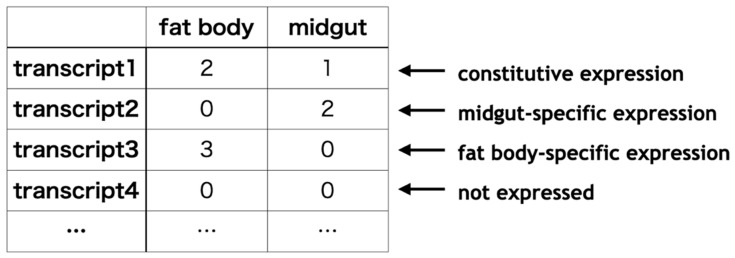
Annotation workflow for transcript description from expression information.

**Figure 4 insects-13-00586-f004:**
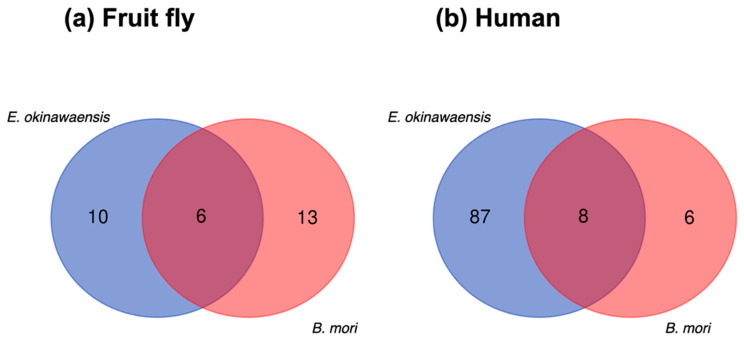
Comparison of ncRNA transcripts between *E. okinawaensis* and *B. mori.* (**a**) The numbers of ncRNA transcripts predicted from fruit fly annotation. (**b**) The numbers of ncRNA transcripts predicted from human annotation.

**Table 1 insects-13-00586-t001:** Source of reference databases.

Category	Name of Resource	URL
Protein sequences	Ensembl	^1^ https://ftp.ensembl.org/pub/release-105/fasta/homo_sapiens/pep/Homo_sapiens.GRCh38.pep.all.fa.gz (accessed on 23 June 2022)
	UniProtKB	https://ftp.uniprot.org/pub/databases/uniprot/current_release/knowledgebase/complete/uniprot_sprot.fasta.gz (accessed on 23 June 2022)
Non-coding RNA sequences	Ensembl	^1^ http://ftp.ensembl.org/pub/release-105/fasta/homo_sapiens/ncrna/Homo_sapiens.GRCh38.ncrna.fa.gz (accessed on 23 June 2022)
	EnsemblGenomes	^1^ http://ftp.ensemblgenomes.org/pub/release-52/metazoa/fasta/drosophila_melanogaster/ncrna/Drosophila_melanogaster.BDGP6.32.ncrna.fa.gz (accessed on 23 June 2022)
Protein and RNA domain	Pfam	http://ftp.ebi.ac.uk/pub/databases/Pfam/releases/Pfam35.0/Pfam-A.hmm.gz (accessed on 23 June 2022)
	Rfam	http://ftp.ebi.ac.uk/pub/databases/Rfam/14.7/Rfam.cm.gz (accessed on 23 June 2022)

^1^ Only the URL for reference data of the typical organism is listed.

**Table 2 insects-13-00586-t002:** Protein-level annotation for *E. okinawaensis*.

Annotation Category	Annotation Level	Number of Annotations	Cumulative Number	Percentage
Protein homolog from tophit	Human or mouse homolog	44,351	44,351	65.1
	*C. elegans* homolog	3739	48,090	70.6
	*D. melanogaster* homolog	2349	50,439	74.0
	Homolog found in UniProtKB	4629	55,068	80.8
No protein homolog	Protein domain	3808	58,876	86.4
	Hypothetical protein	9286	68,162	100

**Table 3 insects-13-00586-t003:** Functional annotation from expression for *E. okinawaensis*.

Annotation Level	All (311,357)	Hypothetical Protein (5699)	Unclassifiable Transcript (253,176)
Fat body-specific expression	48,675	622	42,790
Midgut-specific expression	28,918	315	25,005
Constitutive expression	228,103	4606	181,521
Not expressed	5661	156	3860

**Table 4 insects-13-00586-t004:** Protein-level annotation for *B. mori*.

Annotation Category	Annotation Level	Number of Annotations	Cumulative Number	Percentage
Protein homolog from tophit	Human or mouse homolog	31,354	31,354	68.6
	*C. elegans* homolog	1752	33,106	72.4
	*D. melanogaster* homolog	2113	35,219	77.0
	Homolog found in UniProtKB	2888	38,107	83.4
No protein homolog	Protein domain	5398	43,505	95.2
	Hypothetical protein	2214	45,719	100

**Table 5 insects-13-00586-t005:** Functional annotation of *B. mori* transcripts based on expression.

Annotation Level	All (51,927)	Hypothetical Protein (1486)	Unclassifiable Transcript (12,845)
Fat body-specific expression	39	1	11
Midgut-specific expression	108	0	29
Malpighian tubule-specific expression	83	3	14
Silk gland-specific expression	365	12	92
Testis-specific expression	861	19	492
Ovary-specific expression	179	9	55
Constitutive expression	7825	114	1205
No expression	609	24	125

**Table 6 insects-13-00586-t006:** ncRNA transcripts expressed both in *E. okinawaensis* and *B. mori*.

	Transcript ID	Gene Name	Gene Description
Annotation from fruit fly	FBtr0100888	mt:lrRNA	mitochondrial large ribosomal RNA
	FBtr0345722	asRNA:CR45330	antisense RNA:CR45330
	FBtr0346876	28SrRNA: CR45837	28S ribosomal RNA:CR45837
	FBtr0346877	pre-rRNA:CR45846	ribosomal RNA primary transcript:CR45846
	FBtr0346881	pre-rRNA:CR45847	ribosomal RNA primary transcript:CR45847
	FBtr0346882	18SrRNA: CR45841	18S ribosomal RNA:CR45841
Annotation from human	ENST00000450451		novel transcript
	ENST00000501016		novel transcript
	ENST00000518947	HOXA-AS3	HOXA cluster antisense RNA 3 [Source:HGNC Symbol;Acc:HGNC:43748]
	ENST00000547387		novel transcript, antisense to TUBA1B
	ENST00000618978	U2	U2 spliceosomal RNA [Source:RFAM;Acc:RF00004]
	ENST00000623543		novel transcript, antisense to TUBA8
	ENST00000631211		novel transcript, similar to YY1 associated myogenesis RNA 1 YAM1
	ENST00000638356		novel transcript, antisense to ATP4A

## Data Availability

Workflow developed and original scripts to replicate the study are available on GitHub: https://github.com/bonohu/SAQE/ (accessed on 14 May 2022). The RNA sequencing reads reported in this article are available in the Sequence Read Archive (SRA) under the accession number DRA013458 (DRR346171, DRR346172, and DRR346173). The assembled transcriptome sequences are available in the Transcriptome Shotgun Assembly (TSA) sequence database under the accession number ICSG01(ICSG01000001-ICSG01311357), and the estimated abundance of transcripts is available from the Genomic Expression Archive (GEA) under the accession ID E-GEAD-476.

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
