# Peer review of "Systematic Functional Annotation Workflow for Insects"

_insects, 2022, doi:10.3390/insects13070586_

Round 1
Reviewer 1 Report
The authors developed a systematic functional annotation workflow for insect research termed “Fanflow4Insects”, which enables the automatic annotation of sequences to exploit the exponential increase in genome and transcriptome information. The authors went further to test Fanflow4Insects using the transcriptomes of silkworm (Bombyx mori) and Japanese stick insect (Entoria okinawaensis). The data from Fanflow4Insects can be potentially applicable to the comparative study of insects with distinct phenotypes. Although I am not the expert in sequence annotation field, the story presented in the manuscript flow smoothly and is solidly supported by the data presented. This is a well-drafted manuscript and I only have a few minor suggestions for improvement as indicated below:
Line 143: × g. Delete the space in between.
Line 145: “Purified total RNA (1 μg) samples were” is better to be “Purified total RNA samples (1 μg each) were”.
Line 173: “nematode worm”. Suggested to delete the word “worm” here.
Line 224: “Of 311,357 transcripts, 18.5% of those (57,732 transcripts) could be translated into 68,162 predicted protein sequences with more than 100 amino acids length”. Apparently, the protein number is large than the transcript number. Does it mean some of the transcripts encode more than one protein? If so, please indicate it here to avoid any potential confusion.
Author Response
Response to Reviewer 1
Suggestion 1: The authors developed a systematic functional annotation workflow for insect research termed “Fanflow4Insects”, which enables the automatic annotation of sequences to exploit the exponential increase in genome and transcriptome information. The authors went further to test Fanflow4Insects using the transcriptomes of silkworm (Bombyx mori) and Japanese stick insect (Entoria okinawaensis). The data from Fanflow4Insects can be potentially applicable to the comparative study of insects with distinct phenotypes. Although I am not the expert in sequence annotation field, the story presented in the manuscript flow smoothly and is solidly supported by the data presented. This is a well-drafted manuscript and I only have a few minor suggestions for improvement as indicated below:
Answer 1: Thank you very much for your positive comments for the improvement of our manuscript.
Suggestion 2: Line 143: × g. Delete the space in between.
Answer 2: Thanks for your revision. We revised our manuscript according to your valuable suggestion (13,000×g).
Suggestion 3: Line 145: “Purified total RNA (1 μg) samples were” is better to be “Purified total RNA samples (1 μg each) were”.
Answer 3: Thanks for your revision. We changed our manuscript according to your valuable suggestion.
> Purified total RNA samples (1 μg each)
Suggestion 4: Line 173: “nematode worm”. Suggested to delete the word “worm” here.
Answer 4: Thanks for your revision. We deleted the word “worm” according to your valuable suggestion.
Suggestion 5: Line 224: “Of 311,357 transcripts, 18.5% of those (57,732 transcripts) could be translated into 68,162 predicted protein sequences with more than 100 amino acids length”. Apparently, the protein number is large than the transcript number. Does it mean some of the transcripts encode more than one protein? If so, please indicate it here to avoid any potential confusion.
Answer 5: We revised our manuscript according to your valuable suggestions. For the issue concerning a description about the number of protein sequences in Line 227-231, we added one sentence ‘Some transcripts were predicted to encode more than one protein sequence.’ to avoid any potential confusion. We believed our manuscript had been improved than before.
Reviewer 2 Report
The biological interpretation of sequences of numerous insects remains as a primary bottleneck of transcriptome analysis faced by most scientists engaged in research of insects. To solve this bottleneck problem, the authors have developed a functional annotation workflow, Fanflow4Insects, for insects, and have it validated by using Japanese stick transcriptome as well as transcriptome of Bombyx mori. The Fanflow4Insects provides an novel approch to insight function of insect genes. It is well written in English and displayed clearly.
Author Response
Response to Reviewer 2
Comment: The biological interpretation of sequences of numerous insects remains as a primary bottleneck of transcriptome analysis faced by most scientists engaged in research of insects. To solve this bottleneck problem, the authors have developed a functional annotation workflow, Fanflow4Insects, for insects, and have it validated by using Japanese stick transcriptome as well as transcriptome of Bombyx mori. The Fanflow4Insects provides an novel approch to insight function of insect genes. It is well written in English and displayed clearly.
Answer: Thank you very much for your positive review comments.
Reviewer 3 Report
The manuscript proposes a functional annotation workflow for insect transcriptomes. The manuscript is well written and the illustration of good quality. However, I have few major comments such as the lack of specificity of the workflow toward insects. The only part dedicated to insects is a GGSEARCH with D melanogaster.
The workflow lacks any real innovation in the search of functional annotations as I believe most of bioinformaticians are using the same tools and sometimes more appropriate ones, for instance for functional annotation of ncRNA.
This workflow will not be easy to use by biologist as coding skills are necessary, so there is no novelty for biologist either.
The quality of the transcriptome could also be evaluated with tools such as BUSCO.
The search of ncRNA could be improved using tools specific to short (miRNA) and to long ncRNA (FEELnc), however it will be important for these steps to have information on RNAseq protocols (polyA+, ribosomal depletion, etc)
Do you find the same genes (orthologs) with the fruits fly annotation and the human annotation in the figure 4?
Author Response
Response to Reviewer 3
Suggestion 1: The manuscript proposes a functional annotation workflow for insect transcriptomes. The manuscript is well written and the illustration of good quality. However, I have few major comments such as the lack of specificity of the workflow toward insects. The only part dedicated to insects is a GGSEARCH with D melanogaster.
Answer 1: Thank you very much for your critical review comment. The functional annotation of genes was dramatically improved by using well-annotated species protein sequence datasets (human, mouse, C.elegans and D.melanogaster) compared with the use of only non-redundant dataset (nr), where so many genes tend to be annotated as ‘hypothetical protein’ because gene descriptions of related organisms are frequently assigned. Fanflow4Insects is well tuned for insect species after the use of several insects not only silkworm and Japanese stick insect but also parasitoid wasp Copidosoma floridanum (Ohno et al. 2019 doi: 10.1016/j.ydbio.2019.09.005 and Sakamoto et al. 2020 doi: 10.1186/s12864-020-6559-3). We believe Fanflow4Insects is specially developed workflow for insects. We added these issues in the Discussion section (Line 369-376).
Suggestion 2: The workflow lacks any real innovation in the search of functional annotations as I believe most of bioinformaticians are using the same tools and sometimes more appropriate ones, for instance for functional annotation of ncRNA.
Answer 2: Thank you very much for your critical review comment. A lot of works should be done for what we call ‘functional annotation’. In the current version of Fanflow4Insects, we try to draw functional inference from public databases while the functional annotation of ncRNA should be improved in a coming version of Fanflow4Insects. In addition to the use of GGSEARCH program for protein sequences, we employed CMSCAN program to search against RNA domain database (Rfam) as described in Line 201-205. Moreover, performing functional annotation based on gene expression is challenging (Line 413-435). Finally, there is a need for a reproducible workflow that can be performed on a laboratory PC, including protein sequence similarity searches by global alignment algorithm.
We believe these issues are innovation in insect genomics, and the issues not described in the original manuscript are added in the Discussion section (Line 406-409).
Suggestion 3: This workflow will not be easy to use by biologist as coding skills are necessary, so there is no novelty for biologist either.
Answer 3: Thank you very much for your critical review comment. Because many entomologists study various insect species and require a practical workflow of functional annotation of those, we aimed to develop Fanflow4Insects to be utilized in the laboratory PC mainly for biologists as described in Answer2. Developed openly on GitHub, so it can be easily cloned from GitHub repositories. Also, specific tools including GGSEARCH can easily be introduced through bioconda framework even if users are not a bioinformatician. We added this in the Discussion section (Line 456-458).
Suggestion 4: The quality of the transcriptome could also be evaluated with tools such as BUSCO.
Answer 4: Thank you very much for your suggestion. While the aim of the manuscript was to develop practical workflow to add functional annotation to transcripts, we evaluated the quality of transcriptome sequence sets described in the manuscript.
The result by BUSCO v5.3.2 with transcriptome mode using insecta_odb10 dataset (2020-09-10) with default settings showed 97.1% completeness [43.5% single-copy and 53.6% duplicated] for transcriptomes from fat body and midgut of E. okinawaensis while it showed 93.0 % completeness [47.9% single-copy and 45.1% duplicated] for that only from midgut of E. okinawaensis.
Similar analysis for B. mori trasncriptome by lepidoptera_odb10 dataset (2020-08-05) yields 98.7% completeness [41.0% single-copy, 57.7% duplicated].
We added this in the Result section (Line 227-231 and 267-269).
Suggestion 5: The search of ncRNA could be improved using tools specific to short (miRNA) and to long ncRNA (FEELnc), however it will be important for these steps to have information on RNAseq protocols (polyA+, ribosomal depletion, etc)
Answer 5: Thank you very much for your suggestion. RNAseq used in this study (fat body samples of Japanese stick insect) is typical one as described in Materials and Methods section (Line 137-146). We agree that the information of RNAseq protocols is important for ncRNA studies. The current target of Fanflow4Insects is to annotate functional information from public databases for typical RNAseq reads of insect species. We added this in the Discussion section (Line 409-412).
Suggestion 6: Do you find the same genes (orthologs) with the fruits fly annotation and the human annotation in the figure 4?
Answer 6: Thank you very much for your suggestion. Yes, all E. okinawaensis transcripts annotated and then listed in Table 6 have more than one counterpart transcripts in B. mori by BLASTN search. We added this in the Result section (Line 314-316).
Because so many transcripts correspond to one gene in fruit fly (or human), we are not sure we can call these as ‘orthologs’. We will attach the list of those transcript IDs for E. okinawaensis below.
TRINITY_DN10774_c0_g1_i3
TRINITY_DN1115_c0_g1_i12
TRINITY_DN58674_c0_g1_i2
TRINITY_DN45265_c0_g1_i1
TRINITY_DN1115_c0_g1_i9
TRINITY_DN12652_c0_g2_i1
TRINITY_DN55457_c0_g1_i1
TRINITY_DN1115_c0_g1_i11
TRINITY_DN26769_c0_g1_i6
TRINITY_DN30977_c0_g1_i1
TRINITY_DN578_c2_g1_i1
TRINITY_DN1115_c0_g1_i6
TRINITY_DN100819_c0_g1_i1
TRINITY_DN45265_c0_g1_i2
TRINITY_DN14459_c0_g1_i4
TRINITY_DN26769_c0_g1_i15
TRINITY_DN10774_c0_g2_i1
TRINITY_DN49334_c0_g1_i1
TRINITY_DN87182_c0_g1_i1
TRINITY_DN81701_c0_g1_i1
TRINITY_DN1115_c0_g1_i1
TRINITY_DN44382_c0_g1_i1
TRINITY_DN19121_c0_g1_i1
TRINITY_DN12652_c0_g2_i3
TRINITY_DN26769_c0_g1_i22
TRINITY_DN1115_c0_g1_i10
TRINITY_DN10774_c0_g1_i2
TRINITY_DN1115_c0_g1_i4
TRINITY_DN143975_c0_g1_i1
TRINITY_DN14459_c0_g1_i3
TRINITY_DN116089_c0_g1_i1
TRINITY_DN4310_c0_g1_i6
TRINITY_DN1115_c0_g1_i7
TRINITY_DN10774_c0_g1_i1
TRINITY_DN26769_c0_g1_i8
TRINITY_DN6938_c0_g5_i1
TRINITY_DN26769_c1_g1_i1
TRINITY_DN1115_c0_g1_i8
TRINITY_DN26769_c0_g1_i7
TRINITY_DN26769_c0_g1_i19
TRINITY_DN14459_c0_g1_i5
TRINITY_DN1115_c0_g1_i2
TRINITY_DN66183_c0_g1_i1
TRINITY_DN1115_c0_g1_i16
TRINITY_DN6938_c0_g1_i5
TRINITY_DN15777_c0_g1_i1
TRINITY_DN138831_c0_g1_i1
TRINITY_DN26769_c0_g1_i18
TRINITY_DN14459_c0_g1_i1
TRINITY_DN6938_c0_g1_i2
TRINITY_DN14459_c0_g1_i8
TRINITY_DN43332_c0_g1_i2
TRINITY_DN58674_c0_g1_i1
TRINITY_DN54081_c0_g1_i1
TRINITY_DN26769_c0_g1_i3